# Electrochemical deoxygenative amination of stabilized alkyl radicals from activated alcohols

Jia Xu[1], Yilin Liu[1], Qing Wang[1], Xiangzhang Tao[1], Shengyang Ni[1], Weigang Zhang[1], Lei Yu[1], Yi Pan[1] & Yi Wang [1] ✉

Alkylamine structures represent one of the most functional and widely used in organic synthesis and drug design. However, the general methods for the functionalization of the shielded and deshielded alkyl radicals remain elusive. Here, we report a general deoxygenative amination protocol using alcohol-derived carbazates and nitrobenzene under electrochemical conditions. A range of primary, secondary, and tertiary alkylamines are obtained. This practical procedure can be scaled up through electrochemical continuous flow technique.

Amines are among the most important organic compounds in materials science and pharmaceuticals[1–5], and thus the development of efficient methods for their construction has been a subject of considerable interest to organic chemists. A variety of innovative approaches for alkylamine functionalization have been developed, the most general methods to prepare alkylamines are reductive amination[6], Buchwald–Hartwig amination[7–10], Ullman-type reaction[11,12] and Chan-Lam amination[13]. However, these methods typically rely on anilines as the nitrogen source, which are often prepared via hydrogenation of nitroarenes. Given the potential advantages of nitroarenes as industrial nitrogen feedstocks, their use in the synthesis of alkylamines has attracted significant attention[14–20] In 2015, Baran and coworkers reported a hydroamination approach for synthesizing tertiary alkylamines from nitroarenes and olefins using Fe salts as catalysts and silanes as reductants[21]. Subsequently, similar strategies have been reported by Hu[22], Zhu[23], Radosevich[24], and our group[25], which employ nitroarenes as nitrogen sources and olefins, halides, or boric acid derivatives as alkyl sources. However, these methods are only capable of constructing primary and secondary alkylamines and are not applicable for alkylamines with special substituents such as α-CF₃, α-CF₂H, and benzyl groups. In addition, very rare reports employed the ubiquitous alcohols for efficient deoxygenative conversion[26–33], probably due to the elevated C−O bond energy (BDE = 95 kcal/mol) and the high redox potentials of alkyl alcohols[34]. Specifically, alcohols with distinctive structural features could not deliver the desired amines, such as trifluoromethylated and benzylic substrates, mainly due to the putative intermediate carbon radicals are favored to undergo single-

electron transfer to afford carbanion/carbocation rather than amination. Therefore, the deoxygenative amination of stabilized and activated radicals has remained challenging, and much effort has been devoted to harness those species for further derivatization.

The introduction of a trifluoroethyl group in molecules plays a significant role, as it typically imparts improved pharmacokinetic and pharmacodynamic properties to drug candidates, such as lipophilicity, membrane permeability, and metabolic stability[35,36]. Several drugs featuring a trifluoroethylamine structure have been reported[37,38]. However, the generation of trifluoroethyl radicals requires the corresponding fluorinated alkyl precursors including halides[39], sulfinates[40], and carboxylic acids[41–44] (Fig. 1a). In addition, these approaches are constrained by limited substrate scope, harsh conditions, and expensive catalysts which are not applicable for the amination process.

A diagram of enthalpy shows the relative stability of representative alkyl radicals if the enthalpy of methyl radical was determined as 0 kcal/mol (Fig. 1b). Due to the inductive effect of fluorine atoms, the stability of the fluorinated alkyl radical is expected to be much lower than that of non-fluorinated equivalence (Please see Supplementary Fig. 5 for details). These findings suggest the challenge of trifluoroethyl radical formations from the corresponding alcohols. Furthermore, FMO analysis on the alkyl radical species revealed the considerably increased energy gaps between the SOMO of the fluorinated alkyl radicals and LUMO of nitrosobenzene ($E_{CF3-t-Bu} = 2.72$ eV, $E_{CF2-t-Bu} = 2.42$ eV), compared with the non-fluorinated radical ($E_{t-Bu} = 1.36$ eV) (Fig. 1c). Thus, the fluorinated radicals are tended to protonate rather than reacting with nitrogen sources. Organic

[1]Jiangsu Key Laboratory of Advanced Organic Materials, State Key Laboratory of Coordination Chemistry, School of Chemistry and Chemical Engineering, Nanjing University, Nanjing, China. ✉e-mail: yiwang@nju.edu.cn

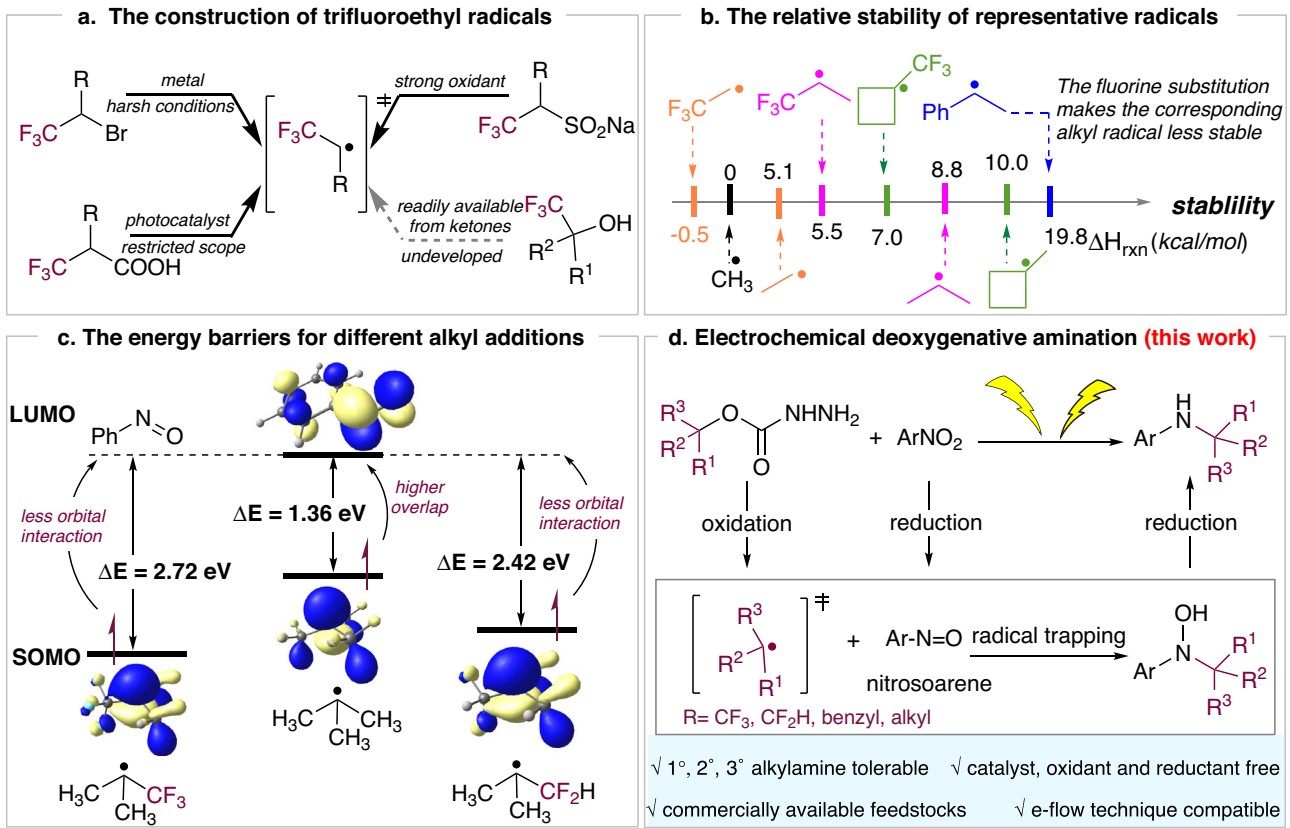

**Fig. 1 | The approaches to fluoroalkylamines. a** The construction of trifluoroethyl radicals. **b** The relative stability of representative radicals. **c** The energy barriers for different alkyl additions. **d** Electrochemical deoxygenative amination.

electrosynthesis, which uses electricity to promote redox transformations, has garnered widespread attention due to its greener and more efficient reaction process[45–51]. Our previous work[52] revealed that the deoxygenation of alcohols can be effectively promoted by installing a carbazate leaving group. This masked deoxidating strategy has been employed to generate primary, secondary, and tertiary alkyl radicals by anodic oxidation. For alcohols with special substituents, we envision that the hydrazine and carboxylate leaving groups could successfully harness the electron-withdrawing effect of the CF₃ group to facilitate the C−O bond cleavage. Thus, these fluorine-containing alkyl carbazate undergoes multiple anodic oxidations to obtain trifluoroethyl radicals with N₂ and CO₂. Meanwhile, the nitroarene as the nitrogen source is reduced at the cathode to produce nitrosoarene synchronously which captures the alkyl radical to form hydroxylamine adduct. Further reduction furnishes the fluoroalkylamine product (Fig. 1d). This paired electrolysis mode differs from traditional electrochemical approaches that involve only a single electrode in the reaction. By simultaneously electrolyzing both the anode and cathode, this method can accommodate redox-neutral reactions rather than being limited to either oxidation or reduction reactions alone.

Herein, we present a general electrochemical paired electrolysis for primary, secondary, and tertiary amines from various functionalized alcohols.

## Results

### Reaction optimization

In our initial investigation, 4-nitrobenzonitrile **1** and 1,1,1-trifluoro-2-methylpropan-2-yl hydrazinecarboxylate **2** were chosen for this deoxygenation amination (Table 1). It was found that graphite as anode and cathode, $^{n}$Bu₄NClO₄ as electrolyte, and N, N-dimethylacetamide (DMA) as solvent in an undivided cell under 10 mA for 6 h at 60 °C, the amination product **3** was isolated in 77% yield (84% GC yield). Using the

Cu or Fe as the cathode could afford lower yields (entries 2–3). Different solvents such as MeCN, DMSO, and DMF were explored and decreased yields were observed (entries 4–6). Using other electrolytes such as LiClO₄ and $^{n}$Bu₄NI led to lower yields of the product in undivided cell (entries 7–8). The reason for the low yield is that nitrobenzene is reduced to aniline, leading to the termination of the reaction. Using 20 mol% ferrocene as oxidant mediate to balance the rate of alkyl production with the reduction rate of nitrobenzene. In addition, product **3** was isolated in a slightly decreased yield of 72% without the 20 mol% ferrocene as oxidant mediate (entry 9). Further control experiments revealed that the amination cannot occur with no electric current (entry 10). The inert atmosphere was crucial for the reaction (entry 11). The reaction efficiency was significantly decreased at room temperature (entry 12). Reducing the amount of carbazate led to a reduced yield of alkylamine (entry 13). In addition, we have tested several plausible pathways for deoxygenative amination. The oxalate and phosphonate[53–55] of trifluoro-t-butanol resulted in trace amount of the product (Please see Supplementary Tables 1–8 for details).

With the optimized reaction conditions, the scope of this deoxygenative amination reaction was explored in Fig. 2. Using 2-trifluoromethyl-2-propanol as alkyl source, 4-chloronitrobenzene, nitronaphthalene, and nitrofluorene was compatible with the transformation, affording the corresponding products in excellent yields (**4**–**6**). The substituent of nitroarenes on meta-position could also tolerate the reaction (**7**). Subsequently, a range of tertiary alkyl carbazates bearing trifluoromethyl were explored for this amination, both linear and cyclic α-trifluoromethyl carbazates were compatible with the transformation, the reaction could proceed with excellent yields (**8**–**13**). When the reaction temperature raised to 80 °C, secondary carbazates such as trifluoroisopropanol and trifluorobenzyl carbazates could generate alkyl radicals smoothly, and corresponding products could be obtained in 33% and 39% yields (**14**–**15**). The low yield is due to

**Table 1 | Optimization of the reaction**

| Entry | Deviation from standard conditions | Yield %[a] |
|---|---|---|
| 1 | None | 84 (77)[b] |
| 2 | C(+)/Cu(−) instead of C(+)/C(−) | 41 |
| 3 | C(+)/Fe(−) instead of C(+)/C(−) | 37 |
| 4 | MeCN instead of DMA | trace |
| 5 | DMSO instead of DMA | 21 |
| 6 | DMF instead of DMA | 72 |
| 7 | $LiClO_4$ instead of $^nBu_4NClO_4$ | 44 |
| 8 | $^nBu_4NI$ instead of $^nBu_4NClO_4$ | 53 |
| 9 | no 20 mol% $Cp_2Fe$ | 72 |
| 10 | no electricity | NR |
| 11 | under air | 24 |
| 12 | 25 °C instead of 60 °C | 53 |
| 13 | 2 equiv. **2** instead of 3 equiv. | 67 |

*NR* no reaction.

[a]Reaction conditions: 4-nitrobenzonitrile **1** (0.2 mmol), carbazate **2** (0.6 mmol), $^nBu_4NClO_4$ (0.2 mmol), $Cp_2Fe$ (0.2 equiv.), DMA (3 mL), graphite anode, graphite cathode, under argon, 60 °C, R = 1000 rpm, 10 mA, 6 h (11.2 F/mol). GC yield using dodecane as internal standard.

[b]isolated yields.

the presence of some by-products of $CO_2$ that have not been removed. Notably, α-difluoromethyl carbazates can also be transformed to generate the corresponding product in 74% yield (**16**).

In addition to fluorinated carbazates, we next turned our focus to explore alkyl carbazates for this amination. A series of primary, secondary, and tertiary alcohol derived carbazates have been treated. The results demonstrated that tertiary alcohol derived carbazates bearing branched (**17**–**24**) and cyclic hydrocarbons (**25**–**30**) were applied to this reaction system to provide the corresponding products in 38–74% yields. It is noteworthy that for the adamantyl alcohol derived carbazate, the corresponding product was obtained with an isolated yield of 71% (**31**). This product is unique and important because the corresponding olefin is not accessible, so existing methods for reductive hydroamination with nitroarenes would not apply. In addition, using α-vinyl tertiary alcohol derived carbazates as deoxygenative reagent, alkyl radical would rearrange, product (**32a**) and rearrangement product (**32b**) were obtained in 30% and 24% isolated yields. Similar to secondary fluorinated carbazates, using primary and secondary carbazates as alkyl sources, the reaction temperature needs to be raised to 80 °C, and the amination products were also synthesized in 17–62% yields (**33**–**47**). Besides, the reaction scope of nitroarenes was investigated (Fig. 3). Nitroarenes bearing both electron-withdrawing groups (F, Cl, Br, I, $CF_3$, CN) and electron-donating groups (iPr, Ph) provided the corresponding products in 47–73% yields (**48**–**55**). Nitronaphthalene (**56**) and nitrofluorene (**57**) were all tolerated for this reaction system. Meta- or ortho-substitution were also suitable reaction partners (**58**–**60**). Subsequently, a variety of nitroarenes including different electronically diverse functional groups were tested with cyclohexanol derived carbazate under 80 °C to afford amination products in good yields (**61**–**67**). Notably, the heterocycle substrates were also suitable substrates, giving the corresponding products in moderate yields (**68**–**72**).

In order to demonstrate the practical value of this paired electrolysis, electrochemical continuous flow technique was successfully incorporated (Fig. 4a). The e-flow system was consisted of specific electrode, base, and gasket (see Supplementary Fig. 2 for details). The reactants were dissolved in the DMA and added into the microchannel reactor through the injection pump. Next, a series of flow rates were screened. Under 0.05–0.1 mL/min flow rates, a considerable amount of the nitroarene was remained with various currents, which may due to short reaction time. Comparison with the reaction in batch, the yield of product **3** was excellent under 0.025 mL/min, and the reaction time was reduced from 6 h to 2 h. In addition, the 10 mmol scale reaction was tested in this e-flow microfluidic system. The product **3** was obtained in 20 h (1.21 g, 71%). These results indicated the practicality of the scaled-up reaction.

To further investigate the mechanism of this electrochemical reaction, a series of control experiments have been carried out (Fig. 4). Under the optimized reaction conditions, by adding radical scavengers 1,1-diphenylethylene could terminate the reaction (Fig. 4b). The radical adduct **73** from 1,1-diphenylethylene was observed on GCMS. Subsequently, different nitrogen sources were examined to verify the possible intermediates (Fig. 4c). Using nitrosobenzene as the nitrogen source, 42% of the amine product was obtained, which indicating that nitrosobenzene might be the reaction intermediate. Moreover, the reaction cannot proceed when nitrobenzene was replaced by aniline. This result indicates that the carbocation pathway was infeasible. N-arylhydroxylamine can be smoothly reduced to aniline, indicating that N-arylhydroxylamine is an intermediate in the reduction process. In addition, the stir rates were tested. Under standard conditions, only trace amounts of product were found without stirring, and at a speed of 1000 rpm, the yield was achieved. Meanwhile, extending the reaction time might lead to a decrease in yield, possibly due to the continuation of the product (Fig. 4d). Subsequently, under the standard condition of no electricity, t-butyl carbazate can generate alkyl radical smoothly using $Fc^+PF_6^-$ as an oxidant (Fig. 4e), and the radical was observed by 1,1-diphenylethylene (**80**). In addition, using nitrosobenzene instead of 1,1-diphenylethylene, intermediate **81** was detected by GCMS. These results shows that ferrocene can be used as an indirect oxidant. Cyclic voltammetry experiments on the reactants were

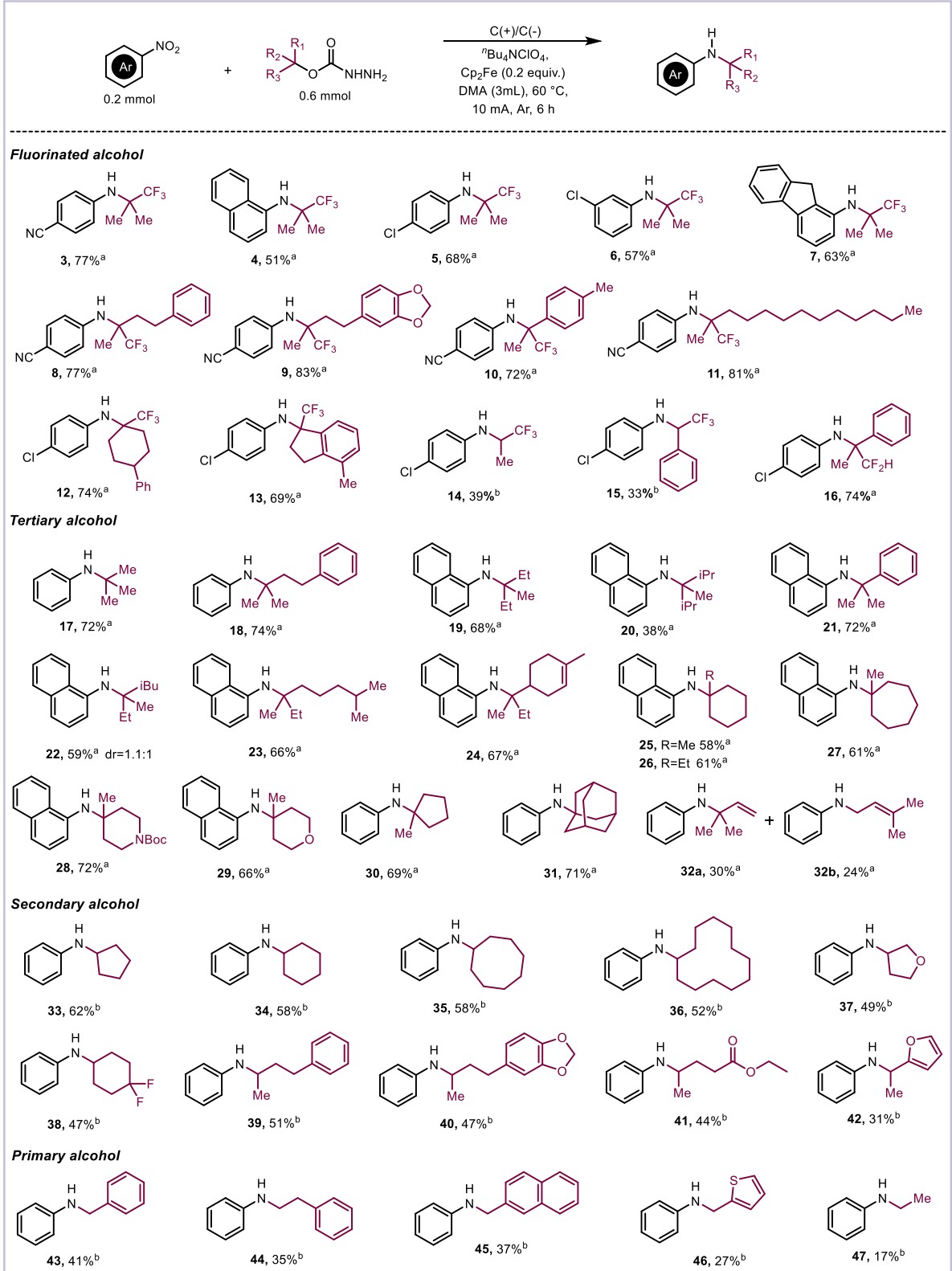

**Fig. 2 | Substrate scope of the carbazates.** [a]Conditions: nitroarenes (0.2 mmol), carbazates (0.6 mmol), [n]Bu$_4$NClO$_4$ (0.2 mmol), Cp$_2$Fe (0.04 mmol), DMA (3 mL), under argon atmosphere, 60 °C, R = 1000 rpm, 6 h. [b]Condition: nitroarenes (0.2 mmol), carbazates (0.6 mmol), [n]Bu$_4$NClO$_4$ (0.2 mmol), Cp$_2$Fe (0.04 mmol), DMA (3 mL), under argon atmosphere, 80 °C, R = 1000 rpm, 6 h.

performed (Fig. 4f). The first oxidation peak of fluoridated t-butyl carbazate **2** was appeared at 0.89 V and the peak of t-butyl carbazate was 0.92 V. The oxidation potential of ferrocene was 0.42 V, which indicated that ferrocene was preferentially oxidized under the cell

conditions to promote further single electron transfer (SET) with carbazate to furnish the diazo radical. Meanwhile, the reduction peak of nitrobenzene with different substituents was texted. 4-nitrobenzonitrile (−0.92 V) was easier to be reduced than

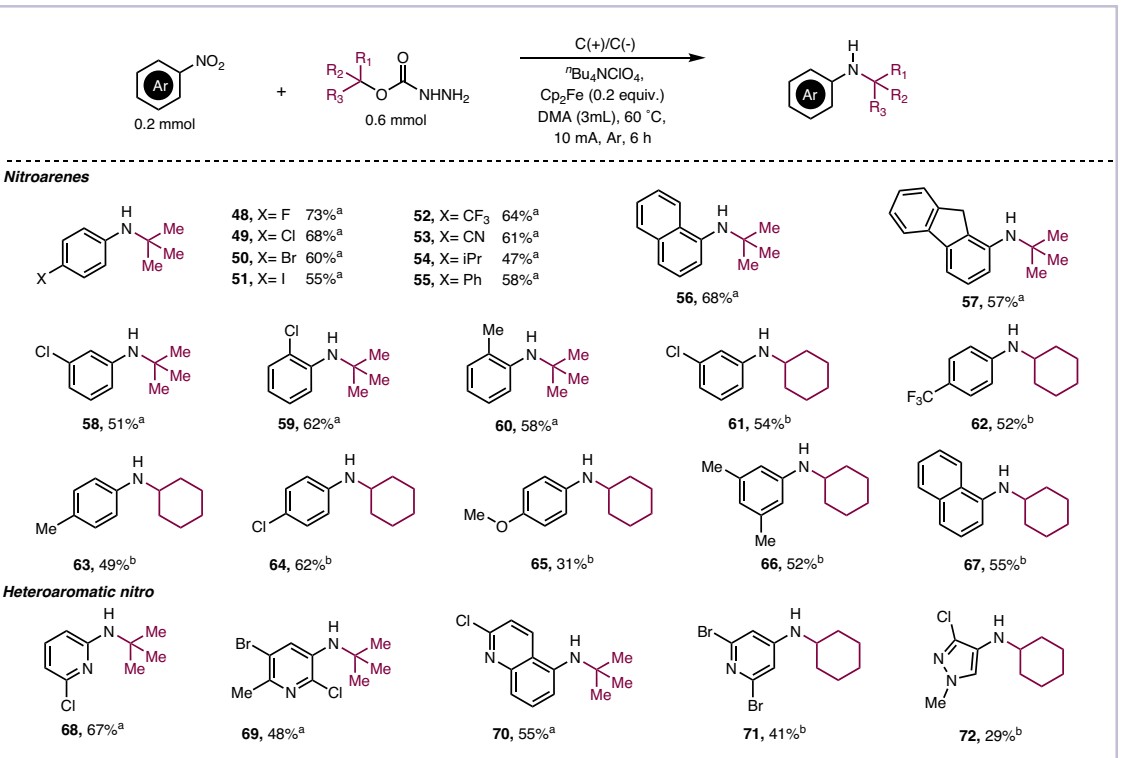

**Fig. 3 | Substrate scope of the carbazates.** [a]Conditions: nitroarenes (0.2 mmol), carbazates (0.6 mmol), [n]Bu4NClO4 (0.2 mmol), Cp2Fe (0.04 mmol), DMA (3 mL), under argon atmosphere, 60 °C, R = 1000 rpm, 6 h. [b]Condition: nitroarene (0.2 mmol), carbazates (0.6 mmol), [n]Bu4NClO4 (0.2 mmol), Cp2Fe (0.04 mmol), DMA (3 mL), under argon atmosphere, 80 °C, R = 1000 rpm, 6 h.

4-nitrocumene (−1.50 V). This measurement result was consistent with the yields of nitrobenzene with different substituents. These results prove that the rate determining step of this reaction is a matching type between the production rate of anodic alkyl radicals and the reduction rate of nitrobenzene. In addition, the reduction peak of nitrosobenzene and N-phenylhydroxylamine were appeared at −1.01 V and −1.45 V, which indicated it gains electron at cathode to promote reaction progression.

Based on the above experiments and previous reports[52,56–60], a plausible reaction mechanism is proposed for the electrochemical transformation at hand (Fig. 5). Initially, the fluorine-containing carbazate **2** was oxidized at anode to generate intermediate **I**. Meanwhile, the part of fluorine-containing carbazate **2**, which need the initial oxidation at the anode involves Cp2Fe (0.42 V), regenerating Cp2Fe⁺. The subsequent oxidation of carbazate **2** by Cp2Fe⁺ through multiple single-electron transfer (SET) and deprotonation processes results in the formation of intermediate **I**. This intermediate subsequently undergoes nitrogen and carbon dioxide liberation, resulting in the generation of an alkyl radical. Simultaneously, nitrobenzene **1** undergoes reduction at the cathode, producing nitrosobenzene **II**, which reacts with the alkyl radical to form oxygen radical **III**. Meanwhile, partial alkyl radical will hydrogenate to produce byproduct. The reduction of **III** leads to the formation of the corresponding anion **IV**, concurrently releasing nitrogen anion **V**. The protonation of **V** culminates in the formation of the desired product **3**.

## Discussion

In summary, we have developed a general deoxygenative C−N coupling reaction of activated alcohols and nitroarenes. Alkyl amines with α-CF3, α-CF2H, and benzyl substituents can be readily accessed under the electrochemical conditions. The mismatched reactivity of alkyl radicals and nitrogen sources has been addressed by paired

electrolysis. The practicality of this method with electrochemical continuous flow technique has been demonstrated.

## Methods
### General information

All commercial reagents were used without additional purification unless otherwise specified. Solvents were purified and dried according to standard methods prior to use. All reactions were run under argon unless otherwise noted. All experiments were monitored by thin layer chromatography (TLC) using UV light as visualizing agent. TLC was performed on pre-coated silica gel plated. Column chromatography was performed using silica gel 60 (300–400 mesh). The instrument for electrolysis is dual display potentiostat (DJS-292B) (made in China). The anode electrode is carbon anode (10 mm × 10 mm × 0.3 mm) and the cathode electrode is platinum plate electrodes (10 mm × 10 mm × 3 mm). ¹H NMR (400 MHz), ¹³C NMR (101 MHz), and ¹⁹F NMR (376 MHz) were measured on a Bruker AVANCE III-400 spectrometer. Chemical shifts are reported in ppm (δ) relative to internal tetramethylsilane (TMS, δ 0.0 ppm) or with the solvent reference relative to TMS employed as the internal standard. Data are reported as follows: chemical shift (multiplicity [singlet (s), doublet (d), triplet (t), quartet (q), broad (br) and multiplet (m)], coupling constants [Hz], integration). Melting points are uncorrected. Infrared spectra were obtained on an Agilent Cary 630 instrument on a diamond plate by way of technology Attenuated Total Reflection (ATR). HRMS were conducted on an Agilent 6540Q-TOF LC/MS equipped with an electrospray ionization (ESI) probe operating in positive ion mode.

### General procedure for synthesis of the α-CF3 carbazates

A round-bottom flask was charged with α-CF3 alcohol (1.0 ml, 10.0 mmol, 1 equiv), followed by the addition of dichloromethane (10 ml) and pyridine (1.5 equiv), The solution was cooled to 0 °C, A

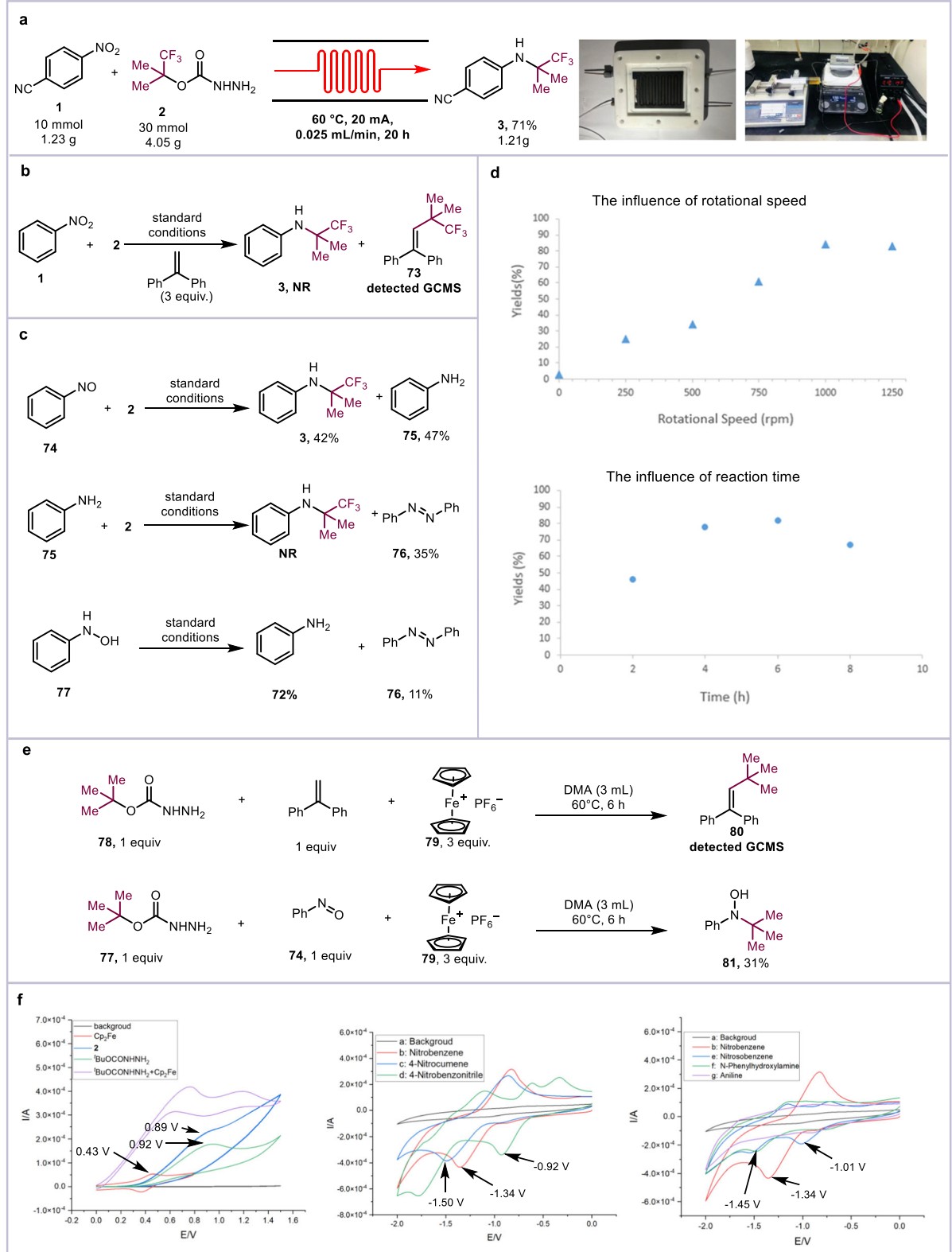

**Fig. 4 | Mechanistic studies and proposal. a** The electrochemical continuous flow system. **b** Radical trapping experiment. **c** The study of reaction intermediates. **d** Controlled experiment. **e** Ferrocenium salt oxidation experiment. **f** Cyclic voltammetry in 0.1 M $^n$Bu$_4$NClO$_4$/DMA with glass carbon working electrode, Pt wire, and Ag/AgCl/KCl (3.0 M) as counter and reference electrode.

solution of phenyl chloroformate (1.38 mL, 11 mmol, 1.1 equiv) in dichloromethane (10 ml) was added. then cooled to room temperature and allowed to stir for overnight. The reaction was quenched with 1 M hydrochloric acid. The aqueous layer was washed with methylene

chloride, dried over Na$_2$SO$_4$, and concentrated in vacuo to afford the crude product carbonate. Next, hydrazine hydrate (2.0 equiv.) was added to the solution of the corresponding carbonate in EtOH (20 ml) and then stirred for about 1 h at 80 °C. Once complete, the reaction

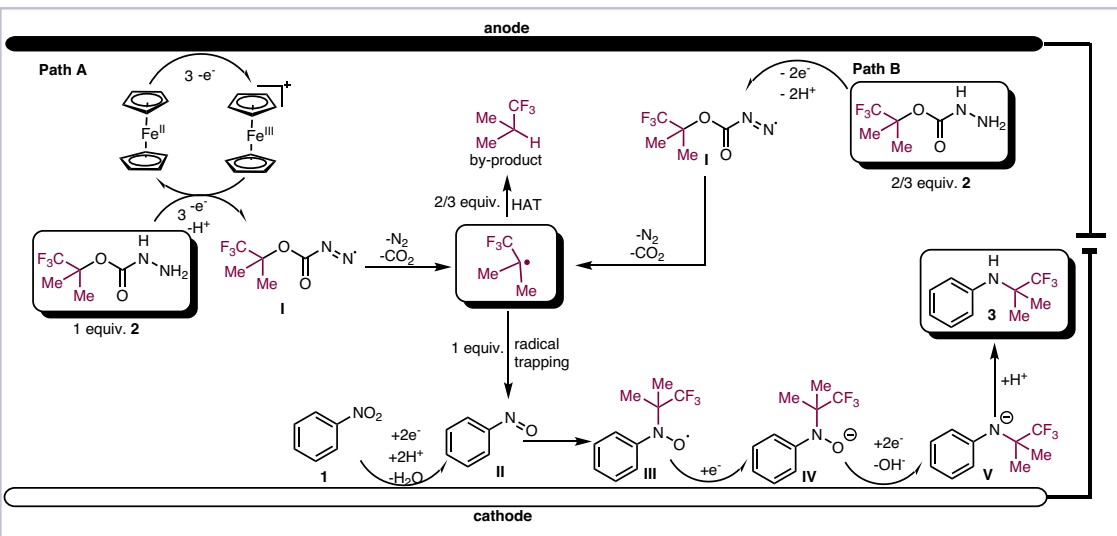

**Fig. 5 | Proposed mechanism.** Two potential reaction pathways involve the oxidation of carbazates, either indirectly via ferrocene or directly through anodic oxidation.

solvent was evaporated under reduced pressure. The corresponding carbazates was purified by silica gel column chromatography.

## Procedure for deoxygenative α-CF3 ammoniation

An undivided bottle was equipped with carbon anode and cathode (10 mm × 10 mm × 3 mm) connected to a DC regulated power supply. To the bottle was added nitrobenzene (26.2 mg, 0.2 mmol), α-CF₃ carbazate (111.6 mg, 0.6 mmol), Cp₂Fe (8.4 mg, 0.04 mmol) tetrabutylammonium perchlorate (68.3 mg, 0.2 mmol) and 3 mL of DMA. The reaction mixture was stirred and electrolyzed at constant current conditions 10 mA at under argon atmosphere (The dual display potentiostat was operating in constant current mode) for 6 h. The reaction was quenched with aqueous NaHCO₃ and extracted with EtOAc (50 mL × 3), the organic solvent was dried over Na₂SO₄. The solvent was evaporated under reduced pressure. The crude product was purified by silica gel column chromatography.

## Data availability

The authors declare that the main data supporting the findings of this study, including experimental procedures and compound characterization, are available within the article and its Supplementary Information files, or from the corresponding author upon request.

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

## Acknowledgements

We gratefully acknowledge the financial support from the National Natural Science Foundation of China Nos. 2201101 and 22271147 for Y.W. We thank Prof. Yong Liang (NJU) and Dr. Hoimin Jung (IBS) for helpful discussions.

## Author contributions

Y.W. designed and guided this project. J.X. is responsible for the plan and implementation of the experimental work. Y.L., Q.W., X.T., and L.Y. analyzed the data. Y.W. and S.N. co-wrote the manuscript. W.Z. and Y.P. discussed the results and commented on the manuscript.

## Competing interests

The authors declare no competing interests.
