## [Peer Review File · Nature Communications]

REVIEWER COMMENTS

Reviewer #1 (Remarks to the Author):

In this manuscript, the authors detail their investigation into the utilization of alcohol-derived carbazates as precursors of alkyl radicals through anodic oxidation. Simultaneously, the reduction of nitroarenes at the cathode leads to the formation of nitrosoarenes. Subsequently, the coupling of the alkyl radical with nitrosoarenes results in the production of the key intermediates for the formation of aniline products. A wide range of nitroarenes and alcohol derivatives including those generated from fluorinated alcohols can be applied in the reaction system, delivering the desired anilines in moderate to good reaction yields. The authors also demonstrated the synthetic utility of this chemistry through the electrochemical continuous flow technique. The possible mechanism for this electrochemical reaction has been investigated.

The application of coupling reactions between alkyl radical species and nitroarenes for the synthesis of anilines is not a new idea (as seen in refs 18-22). The authors have reported a similar approach, employing simple alkanes as radical precursors for the conversion of nitroarenes into anilines (ref 22). Furthermore, the same authors have reported the use of alcohol-derived carbazates as alkyl radical precursors under electrochemical conditions (ref 45). Finally, convergent paired electrolysis of radical intermediates has also been demonstrated in the literature. However, this part of background information has not been discussed in the manuscript. Overall, the concept can be therefore described as the combination of some known methods. The question is now if there is anything exceptional about the practicability and/or utility of this method to appeal to the broad readership of Nature Communications.

The utilization of ubiquitous alcohols as alkylating agents through the cleavage of C-O bonds has garnered significant attention within the synthetic chemistry community. In the current study, carbazates derived from alcohols were synthesized by functionalizing alcohol substrates, which were subsequently reacted with hydrazine hydrate, a hazardous chemical. This preparation method significantly diminishes the practicality of the entire transformation and only very simple alcohols that have no useful functional groups could be employed in this process. The incorporation of fluorine atom in anilines is interesting; however, the potential application of this moiety in medicinal chemistry has not been discussed in the text.

One last comment, the organization of the manuscript and the English language should to be re-checked before submitting anywhere else. The article contains redundant content.

For above reasons, I cannot support the acceptance of the paper for publication in Nature Communications.

Reviewer #2 (Remarks to the Author):

In this manuscript, Wang and co-workers describes the synthesis of N-alkylanilines with α -CF₃, α -CF₂H, and benzyl substituents by using alcohol-derived carbazates with nitrobenzene under paired electrolysis conditions. Hindered amines, in particular α -CF₃, CF₂H substituted amines, are difficult to access. A plethora of methods also exist for the preparation of such arylalkyl-substituted secondary amines. However, this transformation contains multiple oxidation and reduction steps (electron transfer steps), and all those contradictory redox events harmoniously coexist in the same vessel which is indeed difficult to realize by conventional chemistry condition. The broad scope of this transformation is evidenced by its successful application to synthesize over 70 substrates in moderate to good yields. Enriched mechanistic studies are included. This protocol was further scaled up by an electrochemical continuous flow technique which also showed its practicality.

I think this work potentially suitable for publication in the Nature Communications when some additional work is performed as suggested below:

1. There are some writing errors such as capitalization and superscripts in the article, For example, "table 1" in the optimization of conditions, "Fig 3" instead of "Fig 3a", "nBu₄NClO₄" in Figure 4g. In addition, a space is needed between alkylsources.
2. The author used ferrocene as the oxidizing medium in this article. It is inappropriate to mention "without external oxidant or catalyst" in the abstract.
3. The description of compounds 48~72 is repeated in this manuscript.
4. The result of entry 9 in table 1 is inconsistent with the article description. This point needs to be confirmed by the author.
5. The structure of 78, product 3 in Fig 4 is wrong.
6. The author has provided different optimization tables in SI for fluorinated and no fluorinated alcohols, but their reaction conditions are the same. What is the reason for this description?
7. The access to N-tertiary alkylamines via this method, the adamantyl example 31 is unique and important because the corresponding olefin is not accessible, so existing methods for reductive hydroamination with nitroarenes would not apply. This should be reflected in the article.
8. The author explored different nitrobenzenes, and whether the reaction could be extended to nitromethane?
9. Some closely relevant references about the release of molecular nitrogen [Angew. Chem., Int. Ed. 2010, 49,10154; J. Org. Chem. 2014, 79, 446–451; Org. Biomol. Chem.,2022, 20,5704] and electrosynthesis [Green Chem., 2023, 25, 5539; Chin. Chem. Lett., 2023, 34, 108699, Org. Chem. Front., 2023, 10, 3898 and so on] are suggested be cited.

Reviewer #3 (Remarks to the Author):

Comments for Author

In this manuscript, Wang and coworkers describes a paired electrolysis approach to the synthesis of N-alkylanilines starting from nitroarenes and alkylcarbazates. Features an electrochemical method to deoxygenate pre-activated alcohols, creating alkyl radicals which are intercepted by nitro arenes. Cathodic reduction then reveals the secondary amine. A plethora of methods also exist for the preparation of such arylalkyl-substituted amines. However, the synthesis of tertiary alkylamines is rarely reported. The synthetic scope of this reaction is a strength. Especially notable is the access to N-tertiary alkylamines and CF₃, CF₂H substituted amines via this method. In addition, the author explored the mechanism and proposed a reasonable electrochemical mechanism. All the synthesized compounds are well characterized. Considering the novelty and the potential application of this article, the reviewer recommends acceptance of this article by Nature Communications after a minor revision.

- 1) The introduction of this article does not highlight the difficulty of synthesizing tertiary amines, which I believe is an important highlight that needs to be described.
- 2) As depicted in Fig 4g, the authors proposed path A and B for the anodic oxidation of the carbazates to afford intermediate I, since the role of the ferrocene has been suggested to be essential for the reaction, how does the authors quantitatively assign the portion of carbazates ((2/3 equiv. of 2 via path B) in each pathway?
- 3) Why does stirring have such a significant impact on the reaction (Fig 4d). The author needs to explain the reason.
- 4) This reaction tested the reduction potentials of different nitrobenzenes. What is the purpose of this step and which step in the reaction mechanism is the determining one.
- 5) All the tested fluorinated carbazates contain one trifluoromethyl groups, is the protocol applicable to other more challenge substrates with multiple trifluoromethyl groups such as HFIP?
- 6) As shown in Figure 3, the reaction of nitro heteroaromatics with carbazates derived from non-activated alcohols is interesting. Is this protocol compatible with nitro heteroaromatics with trifluoroisopropanol and trifluorobenzyl carbazates?

- 7) Line 68, "...which 68 was more greenly and efficient." should be corrected as "...which is more green and efficient."
- 8) Line 79, "N, N-dimethylacetylacetamide (DMA)" should be "N, N-dimethylacetamide (DMA)".
- 9) Line 87, the yield for entry 9 without ferrocene in Table 1 is 72% rather 56%, please double check it.
- 10) Line 129, the yield for products 44-55 is 47-73% rather than 38%-73%, please double check it.
- 11) Line 137 and 138, "...were also a suitable substrate..." should be corrected as "...were also suitable substrates...".
- 12) Line 139-147, this paragraph is repetitive as the previous paragraph. Please delete this paragraph.
- 13) Line 149, "successful" should be corrected as "successfully".
- 14) Line 187, "texted" is "tested"?
- 15) Fig. 4g, the structure of final product 3 is not correct, please double check it.
- 16) In Reference section, Ref. 2, 3 and 13, "Chemical reviews" should be "Chem. Rev."
- 17) Several writing errors should be corrected. Such as "nBu4NCIO4".

Review comments: major revision

In this manuscript, Wang and co-workers describes the synthesis of N-alkylanilines with α -CF₃, α -CF₂H, and benzyl substituents by using alcohol-derived carbazates with nitrobenzene under paired electrolysis conditions. Hindered amines, in particular α -CF₃, CF₂H substituted amines, are difficult to access. A plethora of methods also exist for the preparation of such arylalkyl-substituted secondary amines. However, this transformation contains multiple oxidation and reduction steps (electron transfer steps), and all those contradictory redox events harmoniously coexist in the same vessel which is indeed difficult to realize by conventional chemistry condition. The broad scope of this transformation is evidenced by its successful application to synthesize over 70 substrates in moderate to good yields. Enriched mechanistic studies are included. This protocol was further scaled up by an electrochemical continuous flow technique which also showed its practicality.

I think this work potentially suitable for publication in the Nature Communications when some additional work is performed as suggested below:

1. There are some writing errors such as capitalization and superscripts in the article, For example, “table 1” in the optimization of conditions, “Fig 3” instead of “Fig 3a”, “nBu4NCIO4” in Figure 4g. In addition, a space is needed between alkylsources.
2. The author used ferrocene as the oxidizing medium in this article. It is inappropriate to mention “without external oxidant or catalyst” in the abstract.
3. The description of compounds 48~72 is repeated in this manuscript.
4. The result of entry 9 in table 1 is inconsistent with the article description. This point needs to be confirmed by the author.
5. The structure of 78, product 3 in Fig 4 is wrong.
6. The author has provided different optimization tables in SI for fluorinated and no fluorinated alcohols, but their reaction conditions are the same. What is the reason for this description?
7. The access to N-tertiary alkylamines via this method, the adamantyl example 31 is unique and important because the corresponding olefin is not accessible, so existing

methods for reductive hydroamination with nitroarenes would not apply. This should be reflected in the article.

8. The author explored different nitrobenzenes, and whether the reaction could be extended to nitromethane?

9. Some closely relevant references about the release of molecular nitrogen [Angew. Chem., Int. Ed. 2010, 49,10154; J. Org. Chem. 2014, 79, 446–451; Org. Biomol. Chem.,2022, 20,5704] and electrosynthesis [Green Chem., 2023, **25**, 5539; Chin. Chem. Lett., 2023, 34, 108699, Org. Chem. Front., 2023, **10**, 3898 and so on] are suggested be cited.

Referee: 1

Comments:

In this manuscript, the authors detail their investigation into the utilization of alcohol-derived carbazates as precursors of alkyl radicals through anodic oxidation. Simultaneously, the reduction of nitroarenes at the cathode leads to the formation of nitrosoarenes. Subsequently, the coupling of the alkyl radical with nitrosoarenes results in the production of the key intermediates for the formation of aniline products. A wide range of nitroarenes and alcohol derivatives including those generated from fluorinated alcohols can be applied in the reaction system, delivering the desired anilines in moderate to good reaction yields. The authors also demonstrated the synthetic utility of this chemistry through the electrochemical continuous flow technique. The possible mechanism for this electrochemical reaction has been investigated.

Response: We appreciate the reviewer for valuable comments and suggestions.

1) The application of coupling reactions between alkyl radical species and nitroarenes for the synthesis of anilines is not a new idea (as seen in refs 18-22). The authors have reported a similar approach, employing simple alkanes as radical precursors for the conversion of nitroarenes into anilines (ref 22). Furthermore, the same authors have reported the use of alcohol-derived carbazates as alkyl radical precursors under electrochemical conditions (ref 45). Finally, convergent paired electrolysis of radical intermediates has also been demonstrated in the literature. However, this part of background information has not been discussed in the manuscript. Overall, the concept can be therefore described as the combination of some known methods. The question is now if there is anything exceptional about the practicability and/or utility of this method to appeal to the broad readership of Nature Communications.

Response: Thank you for your comments. In the past five years, the research on the electrochemical activation of alcohols has attracted the great attention due to their broad application for deoxygenative functionalization. Our group started this hot topic investigation since 2015 and published some interesting papers (*Angew. Chem. Int. Ed.* **2020**, 59, 10859; *Org. Lett.* **2021**, 23, 19, 7524). But to date, the research is only in its infancy with too many important problems to be solved, such as the poor substrate tolerance of tertiary alcohols. Furthermore, the merging electrochemical conditions allow for selective activation of strong C-O bond in the complex molecular skeletons (*J. Am. Chem. Soc.* **2021**, 143, 3536; *Chem. Commun.* **2019**, 55, 15089). However, the deoxygenative amination of tertiary alcohols, especially the challenging fluorine-containing substances have not been reported (*J. Am. Chem. Soc.* DOI: 10.1021/jacs.4c00871). The findings in this submitted manuscript exhibit several significant advances and fill the gap of tertiary alcohol activation in the field of electrochemical amination that render this manuscript suitable for publication.

Novel structural design. Normally, to introduce nitrogen atom into organic molecules, alkyl amines are often constructed by alkyl halides or their surrogates with primary amines. However, those with inadequate sources, especially the tertiary carbon centers are inaccessible, which limited the application for such approaches (*Acc. Chem. Res.* **2008**, 41, 1534; *Chem. Rev.* **2019**, 119, 12491). After more than two years' effort, we finally found the alkyl carbazates that derived from ubiquitous alcohols are also ideal candidates for deoxygenative amination. Benefit from the special redox

properties and incorporated multi-functional structure, the carbazate behaves rather stable on benchtop by extremely active cell conditions. These results confirm that the carbazate derivatives not only generate free radicals but also effectively promotes the following amination process.

Unique electronic property. For the fluorinated alcohols, the obvious inductive effect has made a vast difference for the anodic oxidation for the generation of alkyl radical. The comprehensive computational studies revealed the relative stability of the fluorinated radicals that further influence on the reactivity with nitrosoarene. Therefore, we believe this work provides a blueprint to examine the activated and stabilized alkyl radicals and showcases the technique to handle diverse intermediate species.

Excellent scope and potential chemical application. Fluorinated alcohols have been employed as polar solvents in nucleophilic reactions and electrolytes of lithium batteries. However, those inert liquids have not been employed as fluorinated alkyl feedstock via C-O bond cleavage. In our work, α -CF₃, α -CF₂H and benzyl substituents alcohols can be readily transformed into the corresponding amines under convenient electrochemical conditions, which has potential value in the synthesis of fluorine-containing motifs. Thus, we believe this work represent a breakthrough both in electrochemistry and fluorine chemistry.

Overall, different from our previous reports (*Angew. Chem. Int. Ed.* **2020**, 59, 10859), we have synthesized some fluorinated carbazates which are not reported before. Meanwhile, the common electrochemical reactions nowadays typically involve oxidation or reduction using a single electrode. This is a great waste of energy. Our paired electrolysis reaction is rarely reported in the field of electrochemistry, as it involves the synergistic process of anode and cathode electrodes. We have overcome the difficulty in synthesizing tertiary alkylamines. This work can efficiently synthesize primary, secondary and tertiary amines under mild conditions. This is something that existing methods for synthesizing amines do not possess. Meanwhile, we have synthesized a series of fluorinated tertiary amines. Which are not been reported before.

2) The utilization of ubiquitous alcohols as alkylating agents through the cleavage of C-O bonds has garnered significant attention within the synthetic chemistry community. In the current study, carbazates derived from alcohols were synthesized by functionalizing alcohol substrates, which were subsequently reacted with hydrazine hydrate, a hazardous chemical. This preparation method significantly diminishes the practicality of the entire transformation and only very simple alcohols that have no useful functional groups could be employed in this process.

Response: Thank you for your comments. There have been extensive studies on the use of alcohols as alkyl sources in various transformations (*J. Am. Chem. Soc.* **2021**, 143, 3536-3543; *J. Am. Chem. Soc.* **2023**, 145, 17023-17028; *Org. Lett.* **2022**, 24, 7476-7481). However, these methods are generally applicable to primary and secondary alkyl alcohols, while tertiary alkyl alcohols often fail to react or exhibit extremely low yields. Through literature research, we have found no reports on utilizing activated and unactivated alcohols as alkyl sources for constructing tertiary alkyl amines. We have explored different activation modes of alcohols, such as commonly used oxalates, NHC or PPh₃, but encountered difficulties in amination. The hydrazine group, on the other hand, serves as an activating moiety due to its proven ability to generate radicals easily through anodic oxidation in electrochemical reactions (*Angew. Chem., Int. Ed.* **2010**, 49, 10154; *Green Chem.*, **2023**, 25, 5539;

Chin. Chem. Lett. **2023**, 34, 108699, *Org. Chem. Front.* **2023**, 10, 3898). Consequently, we have chosen to employ the hydrazine group as the activating moiety.

The synthesis of complex alcohols is difficult, and the article does not include examples of drug molecule building blocks. However, we developed this amidation method to address the current challenges in synthesizing tertiary alkyl amines. Our design does not require complex functional groups for validation, as amine compounds are important organic chemicals with numerous applications in the pharmaceutical and materials fields. Any method may have limitations in terms of raw material availability, substrate compatibility, and reaction conditions. From a synthetic perspective, the conversion from a simple alcohol compound to a tertiary alkyl amine, which is not easily obtained, represents a transformation from a low-value-added to a high-value-added chemical product.

3) The incorporation of fluorine atom in anilines is interesting; however, the potential application of this moiety in medicinal chemistry has not been discussed in the text.

Response: Thank you for your comments. We have added a discussion on the potential drug value of trifluoroethylamine compounds in the Introduction. Anilines are structural components of a large and increasing number of bioactive natural and unnatural compounds, the 2,2,2-trifluoroethyl group is very appealing because it usually endows the drug candidates with better pharmacokinetic and pharmacodynamic properties, such as lipophilicity, membrane permeability and metabolic stability. Notable examples of N-trifluoroethylanilines are shown in the below.

4) One last comment, the organization of the manuscript and the English language should to be re-checked before submitting anywhere else. The article contains redundant content.

Response: Thank you for your comments. We have carefully revised the manuscript and made corrections to its English language and writing errors.

Referee: 2

Comments:

In this manuscript, Wang and co-workers describes the synthesis of N-alkylanilines with α -CF₃, α -CF₂H, and benzyl substituents by using alcohol-derived carbazates with nitrobenzene under paired electrolysis conditions. Hindered amines, in particular α -CF₃, CF₂H substituted amines, are difficult to access. A plethora of methods also exist for the preparation of such arylalkyl-substituted secondary amines. However, this transformation contains multiple oxidation and reduction steps (electron transfer steps), and all those contradictory redox events harmoniously coexist in the same vessel which is indeed difficult to realize by conventional chemistry condition. The broad scope of this transformation is evidenced by its successful application to synthesize over 70 substrates in moderate to good yields. Enriched mechanistic studies are included. This protocol was further scaled up by an electrochemical continuous flow technique which also showed its practicality. I think this work potentially suitable for publication in the Nature Communications when some additional work is performed as suggested below.

Response: We appreciate the reviewer for valuable comments and suggestions.

1) There are some writing errors such as capitalization and superscripts in the article, For example, “table 1” in the optimization of conditions, “Fig 3” instead of “Fig 3a”, “nBu4NCIO4” in Figure 4g. In addition, a space is needed between alkylsources.

Response: Thank you for your comments. we have carefully revised the English language. We corrected these writing errors in the manuscript. And these changes were highlighted in yellow in the article.

2) The author used ferrocene as the oxidizing medium in this article. It is inappropriate to mention “without external oxidant or catalyst” in the abstract.

Response: Thank you for your comments. We have corrected “without external oxidant or catalyst” to “without external oxidant or reductant” in the abstract and introduction.

3) The description of compounds 48~72 is repeated in this manuscript.

Response: Thank you for your comments. We are very sorry for a mistake in the manuscript. The duplicate content about the description of compounds 48-72 has been deleted.

4) The result of entry 9 in table 1 is inconsistent with the article description. This point needs to be confirmed by the author.

Response: Thank you for your comments. Under optimal reaction conditions without 20 mol% ferrocene as the oxidant mediate, the product 3 was obtained in 72% GC yield. Using tert-Butyl carbazate as the alkyl source instead of 1,1,1-trifluoro-2-methylpropan-2-yl hydrazinecarboxylate, the corresponding product was obtained in 56% GC yield. We corrected the correct yield in the manuscript.

5) The structure of 78, product 3 in Fig 4 is wrong.

Response: Thank you for your comments. We are very sorry for a mistake in Figure 4e and 4g. We corrected the correct structure in the manuscript.

6) The author has provided different optimization tables in SI for fluorinated and no fluorinated alcohols, but their reaction conditions are the same. What is the reason for this description?

Response: Thank you for your comments. In the process of optimizing the conditions for this reaction. Using tert-butyl carbazate as the alkyl source, the corresponding product was obtained in 56% GC yield. After adding ferrocene, the yield of the reaction increased from 56% to 83%. However, using fluorinated alcohol as alkyl source, the yield only slightly increased from 72% to 84%. Base on the literature (*Chem. Soc. Rev.* **2014**, 43, 2492-2521.). We retested the oxidation potential of **2** and tert-Butyl carbazate. The oxidation potentials of these two are similar (0.89V and 0.92V). The difficulty of this amination reaction is that the rate of generation of alkyl radicals at the anode needs to match the reduction rate of nitrobenzene at the cathode. For tert-butyl carbazate, there is a large amount of aniline present at the end of the reaction without adding Cp_2Fe , tert-Butyl carbazate can produce tert butyl faster through adding Cp_2Fe . For **2**, trifluoromethyl will make the carbon on the C-O bond more electron deficient, making it easier for oxygen to leave. We believed that this phenomenon was important, and provided different optimization tables in SI for fluorinated and no fluorinated alcohols.

7) The access to N-tertiary alkylamines via this method, the adamantyl example **31** is unique and important because the corresponding olefin is not accessible, so existing methods for reductive hydroamination with nitroarenes would not apply. This should be reflected in the article.

Response: Thank you for your comments. Based on the previous reports, the N-tertiary alkylamines with the adamantyl was difficult to synthesize. In 2015, Baran and coworkers reported a hydroamination approach for synthesizing tertiary alkylamines (*Science* **2015**, 348, 886-891). This

method uses olefins as alkyl sources and is not suitable for adamantanes. We highlighted compound **31** in the manuscript.

8) The author explored different nitrobenzenes, and whether the reaction could be extended to nitromethane?

Response: Thank you for your comments. Under standard conditions, we used nitromethane as the amine source, but unfortunately no target product was generated, where alkyl radicals underwent hydrogenation rather than being captured by nitroso methane. Through literature research (*Nat. Commun.* **2016**, 7, 12494; *J. Am. Chem. Soc.* **2020**, 142, 16205-16210), we found that the activity of nitromethane was significantly lower than that of nitrobenzene compounds. The slow reduction rate of nitromethane made it unsuitable for this reaction.

9) Some closely relevant references about the release of molecular nitrogen [*Angew. Chem., Int. Ed.* **2010**, 49,10154; *J. Org. Chem.* **2014**, 79, 446–451; *Org. Biomol. Chem.*,**2022**, 20,5704] and electrosynthesis [*Green Chem.*, **2023**, 25, 5539; *Chin. Chem. Lett.*, **2023**, 34, 108699, *Org. Chem. Front.*, **2023**, 10, 3898 and so on] are suggested be cited.

Response: Thank you for your comments. These closely relevant references about the release of molecular nitrogen have been cited in the manuscript.

Referee: 3

Comments:

In this manuscript, Wang and coworkers describes a paired electrolysis approach to the synthesis of N-alkylanilines starting from nitroarenes and alkylcarbazates. Features an electrochemical method to deoxygenate pre-activated alcohols, creating alkyl radicals which are intercepted by nitro arenes. Cathodic reduction then reveals the secondary amine. A plethora of methods also exist for the preparation of such arylalkyl-substituted amines. However, the synthesis of tertiary alkylamines is rarely reported. The synthetic scope of this reaction is a strength. Especially notable is the access to N-tertiary alkylamines and CF₃, CF₂H substituted amines via this method. In addition, the author explored the mechanism and proposed a reasonable electrochemical mechanism. All the synthesized compounds are well characterized. Considering the novelty and the potential application of this article, the reviewer recommends acceptance of this article by Nature Communications after a minor revision.

Response: We appreciate the reviewer for valuable comments and suggestions.

1) The introduction of this article does not highlight the difficulty of synthesizing tertiary amines, which I believe is an important highlight that needs to be described.

Response: Thank you for your comments. We have elucidated the challenges in the synthesis of tertiary amines and described our methods to overcome these difficulties in the introduction.

2) As depicted in Fig 4g, the authors proposed path A and B for the anodic oxidation of the carbazates to afford intermediate I, since the role of the ferrocene has been suggested to be essential for the reaction, how does the authors quantitatively assign the portion of carbazates ((2/3 equiv. of 2 via path B) in each pathway?

Response: Thank you for your comments. Paired electrolysis necessitates an equivalent coulometric count at both the anode and the cathode. In this reaction, there is a transfer of three electrons at the anode and five electrons at the cathode. According to the law of charge conservation, the anode preferentially converts ferrocene into alkyl radicals, whereas the carbazates undergo direct oxidation to yield alkyl radicals. Experimental data further suggest that multiple equivalents of carbazates are required. Upon completion of the reaction, gas chromatography-mass spectrometry (GC-MS) analysis reveals that the excess alkyl radicals undergo hydrogenation.

3) Why does stirring have such a significant impact on the reaction (Fig 4d). The author needs to explain the reason.

Response: Thank you for your comments. We tested at different speeds and found that without stirring, the reaction only produced trace amounts of products. Nitrobenzene is mainly reduced to aniline. Increasing the speed will significantly increase the yield, reaching maximum yield at 1000 rpm. Anodic oxidation of the carbazate generates alkyl radical, which intercepts a cathodically generated nitrosobenzene and ultimately gives the final product upon subsequent reduction. When the reaction is not stirred, the alkyl radicals generated by the anode cannot add nitrosobenzene produced by the cathode. Nitrosobenzene will be further reduced to aniline, causing the reaction to stop.

4) This reaction tested the reduction potentials of different nitrobenzenes. What is the purpose of this step and which step in the reaction mechanism is the determining one.

Response: Thank you for your comments. The reduction peak of nitrobenzene with different substituents were tested. 4-nitrobenzotrile (-0.92 V) was easier to be reduced than 4-nitrocumene (-1.50 V). This measurement result was consistent with the yields of nitrobenzene with different substituents. These results prove that the rate determining step of this reaction is a matching type between the production rate of anodic alkyl radicals and the reduction rate of nitrobenzene. This indicates that a faster rate of alkyl radical generation is required, with more uniform stirring, so that nitroso benzene can immediately capture alkyl radicals.

5) All the tested fluorinated carbazates contain one trifluoromethyl groups, is the protocol applicable to other more challenge substrates with multiple trifluoromethyl groups such as HFIP?

Response: Thank you for your comments. We tried to synthesize carbazates contain HFIP. As shown in the figure below, unfortunately, after adding hydrazine hydrate, intermediate 1,1,1,3,3,3-hexafluoropropan-2-yl phenyl carbonate will hydrolyze back to HFIP instead of forming carbazate.

6) As shown in Figure 3, the reaction of nitro heteroaromatics with carbazates derived from non-activated alcohols is interesting. Is this protocol compatible with nitro heteroaromatics with trifluoroisopropanol and trifluorobenzyl carbazates?

Response: Thank you for your comments. As illustrated in the figure below, we tested 2-chloro-6-nitropyridine as an amine source under standard conditions, resulting in the target product of 32% isolated yield. Unfortunately, during the isolation process, the product was found to be unstable, precluding the acquisition of accurate NMR spectra. Consequently, examples containing nitro groups are not included in the manuscript.

7) Line 68, "...which 68 was more greenly and efficient." should be corrected as "...which is more greenly and efficient."

Response: Thank you for your comments. "...which 68 was more greenly and efficient." was corrected as "...which is more greenly and efficient." in the manuscript.

8) Line 79, “N, N-dimethylacetylacetamide (DMA)” should be “N, N-dimethylacetamide (DMA)”.

Response: Thank you for your comments. “N, N-dimethylacetylacetamide (DMA)” was corrected as “N, N-dimethylacetamide (DMA)”.

9) Line 87, the yield for entry 9 without ferrocene in Table 1 is 72% rather 56%, please double check it.

Response: Thank you for your comments. Under optimal reaction conditions without 20 mol% ferrocene as the oxidant mediate, the product 3 was obtained in 72% GC yield. Using tert-Butyl carbazate as the alkyl source instead of 1,1,1-trifluoro-2-methylpropan-2-yl hydrazinecarboxylate, the corresponding product was obtained in 56% GC yield. We corrected the correct yield in the manuscript.

10) Line 129, the yield for products 44-55 is 47-73% rather than 38%-73%, please double check it.

Response: Thank you for your comments. We are very sorry for this mistake, the yields for products 44-55 were corrected as 47-73%.

11) Line 137 and 138, “...were also a suitable substrate...” should be corrected as “...were also suitable substrates...”.

Response: Thank you for your comments. “...were also a suitable substrate...” was corrected as “...were also suitable substrates...”.

12) Line 139-147, this paragraph is repetitive as the previous paragraph. Please delete this paragraph.

Response: Thank you for your comments. We are very sorry for a mistake in the manuscript. The duplicate content about the description of compounds 48-72 has been deleted.

13) Line 149, “successful” should be corrected as “successfully”.

Response: Thank you for your comments. “successful” was corrected as “successfully”.

14) Line 187, “texted” is “tested”?

Response: Thank you for your comments. “texted” was corrected as “tested”.

15) Fig. 4g, the structure of final product 3 is not correct, please double check it.

Response: Thank you for your comments. We corrected the structure in figure 4g.

16) In Reference section, Ref. 2, 3 and 13, “Chemical reviews” should be “Chem. Rev.”

Response: Thank you for your comments. “Chemical reviews” in reference section was corrected as “Chem. Rev.”

17) Several writing errors should be corrected. Such as “nBu4NClO4”.

.

Response: Thank you for your comments. These writing errors were corrected in this manuscript.

REVIEWERS' COMMENTS

Reviewer #2 (Remarks to the Author):

The author addressed most of my concerns, and I recommended publication in the Nature Communications.

Reviewer #3 (Remarks to the Author):

After revision, my comments have been mostly addressed, thus I recommend acceptance for this manuscript.